# Gama rays mediated improvement of catalytic efficiency and thermostability of glucoamylase by replacing active site leucine to isoleucene from super koji (*Aspergillus oryzae*)

**Anam Saqib**[1,2]**, Saif -ur-Rehman**[1]**, Hazrat Ali**[1]***, Noor Hassan**[1]**, Asad Ali**[1]**,
**Muhammad Hamid Rashid**[1]***

**1** National Institute for Biotechnology and Genetic Engineering (NIBGE)—College, Pakistan Institute of Engineering and Applied Sciences (PIEAS), Islamabad, Pakistan, **2** QuinTech Center for Applied Sciences (QCAS), Lahore, Pakistan

* hamidcomboh@gmail.com (MHR); alibiotechnologist@gmail.com (HR)

## Abstract

Glucoamylase is considered as an essential enzyme in food industry. However, lowere catalytic efficiency and weak thermostability confine its application in food industry. Therefore, the current study was aimed to improve catalytic efficiency and thermostability of glucoamylase by replacing active site leucine to isoleucene from Super Koji (*Aspergillus oryzae*) using gama rays mediated point mutation. High catalytic efficiency and thermostability of glucoamylase from mutant *Aspergillus oryzae* M-60(5) (screened from 51 mutants) was achieved due to a point mutation, i.e., Leu203→Ile in active site. The SDS-PAGE molecular mass of parent and mutant glucoamylase was 63.1 kDa, while mutant glucoamylase showed; productivity = 9.7 U ml$^{-1}$, kinetic constants $k_{cat}$ = 118 (1.62 fold), ($k_{cat}/K_m$) = 1899 (4.75 fold) and half-life at 55 °C for 45 min (1.92 fold). Thermodynamics parameters for starch hydrolysis of parent glucoamylase were; $\Delta H^*$ = 47.755 kJ mol$^{-1}$ and $\Delta G^*$ = 67.975 kJ mol$^{-1}$ while for mutant $\Delta H^*$ = 44.263kJ mol$^{-1}$ and $\Delta G^*$ = 66.514 kJ mol$^{-1}$. The $\Delta G^*$ of irreversible thermostability for parent and mutant at 55 °C was 104.95 kJ mol$^{-1}$ and 101.52 kJ mol$^{-1}$ respectively. The point mutation altered the conformation of the glucoamylase active site that contributed to improve the functional energy ($\Delta G^*$), resulted the stabilization of transition state which made it thermostable and highly efficient in starch hydrolysis.

## Introduction

Glucoamylase (GA) is the most widely used enzyme [1] and has distinctive properties of hydrolyzing starch into subunits of oligosaccharide and commonly used for production of high corn, glucose and fructose syrups as well as for alcohol production [2–4]. The GA produces glucose directly by scarification of α-1,3, α-1,4 and α-1,6 glycosidic bonds present at non-reducing end of starch molecule [5,6]. *Aspergillus oryzae* is the main focus of food

**Data availability statement:** All relevant data are within the manuscript.

**Funding:** The author(s) received no specific funding for this work.

**Competing interests:** The authors have declared that no competing interests exist.

industries for hydrolytic enzymes production such as α-amylase, GA and other food grade enzymes. Different fungal species, e.g., *Aspergillus niger*, *Rhizopus niveus* and *Rhizopus delemar*, are used to produce GA commercially [7]. However, their lower catalytic activity leads to consume high energy and restrictions in starch processing and fermentation [8]. Hence, it is a dire need to develop novel fungal GA suitable for evolving industrial applications.

The *A. oryzae* is considered as Generally Recognized As Safe (GRAS), therefore their enzymes are safe for food and feed industry [8,9]. In recent era, strain improvement by site directed and random mutagenesis of fungi is being used widely for industrial applications. Strain improvement usually targets for elevated yields of the enzymes and biomass production, broad spectrum uses of industrially suitable substrates and efficient physiological properties [10]. Implementation of various molecular methods like recombinant DNA technology and γ-rays dependent mutagenesis is being carried out to evolve strains in order to meet increase industrial demands for food grade enzymes [11].

The complete open reading frame of *A. oryzae* GA contained 612 amino acids residues and GA protein is composed of carbohydrate binding module (CBM) of 106 amino acid residues from 506–612 [12]. The *A. oryzae* GA showed 67% and 30% homology with *A. niger* and *R oryzae*, respectively [13,14]. Five highly conserved regions were reported in GA enzyme of *A oryzae* and *A. niger* with 78–95% similarity, whereas, the active site of *A. oryzae* was homologous to *A. niger* [15].

Random mutagenesis induced by γ-rays may contribute in enhancement of industrial enzymes production; hence γ-ray mediated mutagenesis of *A. niger* made the mutant GA highly efficient in substrate hydrolysis (Aleem et al. 2018). Kinetics and thermodynamics of GA suggested that the mutated enzyme have potential for commercial scale glucose production in starch processing as well as in the food industry [16,17]. Previously, we developed novel *Aspergillus oryzae* mutants by γ-rays' mutagenesis for the enhanced production of thermostable α-amylases, which were also highly specific in the α-amylase production [18]. The increase in production, catalytic efficiency and thermostability of α-amylases proved that the γ-rays might have altered the α-amylases conformation [18]. In the current study, we further have screened the previously generated koji mutants for the thermostable GA hyper producer strains. Consequently, mutant M-60(5) was identified as the potent hyper producer of thermostable GA.

Novelty of current report is as it for the first time explains about effect of point mutation, i.e., replacement of catalytic center leucine to iso-leucine on the active site conformation; kinetics and thermodynamics of stability-function of the GA. Moreover, mutation's effect on the GA was further attributed through evaluation of active site microenvironment by determining the heat of ionization of active site residues.

## Methods

### Culture maintenance and mutant preparation

The super Koji (*A. oryzae* cmc1) and its mutant derivative M-60(5) strain (hyper producer of thermostable GA) were obtained from Industrial Enzymes & Biofuels Group, Industrial Biotechnology Division, National Institute for Biotechnology and Genetic Engineering (NIBGE), Faisalabad. The Koji M-60(5) mutant was screened from the mutants of super Koji, which were previously developed by γ-rays treatment [18]. Briefly, the mutants were generated by preparing fresh inoculum of *A. oryzae* in six 15 ml falcon tubes and irradiated with Caesium-137 (Cs-137) γ-ray and γ-rays source fitted in gamma cell radiation chamber (Mark-IV Irradiator/Gamma Cell-220). Main stock of mutants (51 variants) was exposed to various level of γ-ray exposure, e.g., 60, 80, 100, 120 & 140 kRad (0.6, 0.8, 1.0, 1.2 & 1.4 kGray). The potato dextrose agar was used to maintain culture at 30 °C as described by Aleem et al. [18].

## Production and purification of GA

The GA from parent and mutant M-60(5) strains was produced in 10L bioreactor. Briefly, six liter of liquid fungal growth medium (LFGM) containing Rafhan starch (2% w/v) with pH 6.5 was prepared. The media was autoclaved for 40 min at 121 °C, 30 psi. Afterwards, the temperature was lowered down to 30 °C and stirred continuously at 150 rpm. The inoculum was grown in 500 ml flask as described [18]. The cell density of inoculum in triplicate was measured and transferred to bioreactor aseptically (0.3% w/v pack cells). The fermenter was run for 72 hrs and aliquots were taken from the bioreactor after every 6 hrs and centrifuged ($25,900 \times g$) for 10 mints to clear the sample. Afterwards, the total cell mass and GA activity was determined in the samples. The muslin cloth (maximum pore size 2 mm) was used to filter growth medium and the filtrate was centrifuged ($25,900 \times g$) for 20 min at 4 °C. Finally, the supernatant was lyophilized using Benchtop Freeze Dryers, (Labconco™ FreeZone™, 115V US Models) to concentrate the enzyme as described [19].

The GA was purified by fractional precipitation. Briefly, the solid ammonium sulphate was added steadily to 1.0 ml of concentrated crude GA to achieve saturation ranging from 10–90% at 4 °C. The GA preparations containing ammonium sulfate were then placed overnight at 4 °C and centrifuged ($25,900 \times g$) for 20 min. The pallet was discarded and GA activity was checked in the supernatants. Salt concentrations relating to onset and the complete precipitation of GA were selected. Hence, for large scale purification of GA, solid ammonium sulfate was added slowly to achieve 40% saturation and placed overnight at 4 °C and centrifuged as described above. After that pellet was discarded and 75% saturation was achieved by adding more salt. Again, supernatant was discarded and pellet containing GA was kept for next step. Distilled water was added to pellet and dialyzed for removing salt at 4 °C for 24 hrs.

## Sequencing and *in-silico* point mutation identification in GA gene

The freshly grown mycelia of parent and mutant strain M-60(5) were used for genomic DNA extraction, which was extracted as reported [20]. The PCR fragments were obtained by using 5′ATGCGGAACAACCTTCTTTTTTCC3′ and 5′CTACCACGACCCAACAGTTGGG3′ as primers, according to thermo profile of 94 °C for 5 min; 35 cycles of 94 °C for 60 s; 61 °C for 1 min, 30 s; 72 °C for 2 min and 72 °C for 10 min. The amplified DNA fragment was applied on poly acrylamide gel for the gene identification and PCR product was purified by using PCR purification kit (GeneJET PCR Purification Kit, Catalog number: K0701, Fermentas, Thermoscitific®) and sequenced by Macrogen, Republic of Korea.

The nucleotide sequence alignment of mutant M-60(5) was performed by Clone Manager 10 tool [21] against parent koji strain. The local and global protein alignment tools at European Molecular Biology Open Software Suite (EMBOSS) was used for converting cDNA to amino acid [22]. The InterPro at European Molecular Biology Laboratory- European Bioinformatics Institute (EMBL-EBI) and National Center for Biotechnology Information (NCBI) conserved domain database were utilized for deducing mutant domain information [23]. After domain localization, 3D structure was constructed by Swiss modeling [24] and superimposing of parent and mutant M-60(5) structures was done by PyMol [25] to observe mutational change in active site of the GA encoding protein.

## Glucoamylase (GA) assay

The GA activity was determined using soluble starch (1% w/v) as substrate. The assay reaction consisting 100 μl enzyme extract, 1.0 ml of soluble starch (1% w/v) in sodium acetate buffer pH 5.0 and incubated at 50 °C for 40 min. The quenched reaction mixture (QRM) in 1.0 ml of Glucose oxidase/Peroxidase kit (Fluitest® GLU, Biocon, Bangalore, India) was used to

determine released glucose [20,26]. One unit of GA activity was defined as the amount of GA required to release one µmol of glucose min$^{-1}$ from soluble starch under defined assay conditions of temperature and pH. The GA activity units were calculated by using the formula as described by Aleem et al. [18].

## Protein assay

The extracellular proteins released were estimated by using Bradford assay and bovine serum albumin was used as a standard [27].

## Molecular mass determination

The purity of purified GA and their subunit molecular mass was determined by 10% sodium dodecyl sulphate denaturing-renaturing polyacrylamide gel electrophoresis (SDS-DR-PAGE). Protein markers from Thermoscientific® ranging in size between 10–180 kDa were used and run as standard. Coomassie brilliant blue R-250 solution (0.1%) was used to stain gel containing enzyme and molecular markers. Apparently pure GAs (0.5 µL of 0.6 mg mL$^{-1}$) was spotted separately on SDS-PAGE gel and incubated at 50 °C for 90. Later, replica copy of the SDS-PAGE gel was cut and stained with Iodine solution [28]. A transparent band appeared after 20 min of staining with blue back ground.

## Optimum pH, pKa & heat of ionization (ΔHI) of active site residues

The purified GA extracted from mutant M-60(5) was checked against various pH using different buffer systems, e.g., Citrate buffer: pH 3.0–6.2, Sorenson's buffer: pH 5.8–8.0 & Glycine-NaOH buffer: pH 8.6–10.6. Variable temperatures (40–55 °C) were used to find the optimum pH of GA. In addition, Dixon and Webb [29] protocol was used to determine the dissociation constants ($pKa_1$ and $pKa_2$) of active site ionizable residues of GA forming ES-complex. Finally, the $\Delta H_I$ was calculated by using the equation below:

$$\Delta H_I = -\text{slope} \times 2.303\text{R}.$$

## Temperature optimum, temperature quotient ($Q_{10}$) & activation energy

GA activity was checked against different temperature ranging from 45–60 °C. The assay was conducted at pH 5 (50 mM Na-acetate buffer) for 45 min. The activation energy ($E_a$) was calculated using Arrhenius plot, while temperature quotient ($Q_{10}$) was calculated as described [29].

$$Q_{10} = \text{antilog}_\varepsilon \ (E \times \ 10 \ / \ \text{RT}^2)$$

$$E = E_a = \text{activation energy}$$

## Kinetics & thermodynamics of starch hydrolysis

Different defined amount of GA with various conc. of substrate (soluble starch) ranging from 0.025% to 0.25% (w/v) were used to determine the Michaelis Menten kinetic constants ($K_m$, $V_{max}$, $k_{cat}$ and $k_{cat}/K_m$) for soluble starch hydrolysis by Koji's GAs at 50 °C, pH 5. The graphpad prism version 7.04 was used to fit date to non-linear regression. Furthermore, Eyring's absolute rate equation was used to calculate the thermodynamic parameters for substrate hydrolysis usingformula described by Eyring and Stearn [30].

$$k_{cat} = \left( k_b T / h \right) e^{(-\Delta H^* / RT)} \cdot e^{(\Delta S^* / R)}$$

## Thermodynamics of irreversible thermal stability

To determine irreversible thermal inactivation of Koji's GAs 15 ml of the parental and mutated GA solutions were taken in falcon tubes and incubated at various temperatures like: 45, 50, 55 and 60 °C in a water bath. The GA aliquots were taken from each sample after regular time intervals, i.e., 0, 5, 10, 15, 20, 25, 30, 35, 40, 45 & 50 min, which were then placed in ice cold water for 30 min for the refolding. Afterwards, the withdrawn GA aliquots were assayed for % residual activity of the GA. The data was fitted to first order plot and rate constants for irreversible thermal inactivation ($K_d$) of GAs were calculated.

The activation energy for irreversible thermal inactivation [$E_{a(d)}$] of GA was determined by using the Arrhenius plot. Rearranged Erying's absolute rate equation was applied to determine the thermodynamics of irreversible inactivation of GAs as mentioned above with the difference that $k_{cat}$ was replaced with $K_d$ (denaturation constant). Moreover, in entalphy change ($\Delta H^*$) determination the $E_a$ was replaced with $E_{a(d)}$.

## Toxin analysis by Liquid chromatography–mass spectrometry

The sample was prperaed by taking equal volume of mutant GAs and 100% chloroform and mixed in a separating funnel. The chloroform was evaporated in rotary evaporator and 2 ml methanol was used to disolve the residues. After that, the mixture was filtered through 0.45 µm nylon membrane and subjcted to analysis by Quadrupole Linear Ion Trap Mass Spectrometer Finnegan LTQ XL hyphenated with Surveyor Plus LC system (LC-MS) (Thermo Fisher Scientific, USA) using the protocal described by [18] and [31] with slight changes. The parameter were set as capillary temperature 335 °C, voltage 45 V, spray voltage 5 kV, sheath gas flow rate 70 and auxiliary gas flow rate 20 arbitrary units. Different aflatoxins ml$^{-1}$ (B1, B2, G1 and G2) were used as standards.

# Results

## Production and purification of GA

In the current study, parent super Koji (*A. oryzae*) and its mutant derivative M-60(5) strain were screened based on the hyper production of thermostable GA were grown under submerged conditions on raw maize starch in 10L fermenter. The GA produced by mutant M-60(5) was 9.7 U ml$^{-1}$ which was 2.6 fold higher than the parent (3.6 U ml$^{-1}$). Whereas, the specific activity of the mutant was 1.83 fold increased (54.9 U mg$^{-1}$) as compare to parent. The GA produced by super koji parent and mutant strains was subjected to single step purification. Fractional precipitation of GAs by ammonium sulfate gave single band on 10% SDS-PAGE, which confirmed the GAs had purity apparently at homogeneity level. The precipitation trend of the mutant enzyme was slightly faster than the control depicting that the γ-rays might has changed the surface of the glubular protein. After purification, the specific activity of the purified GAs from parent and mutant M-60(5) Koji strains was increased to 51.8 and 96.2 U mg$^{-1}$, respectively.

## Sequencing and point mutation identification in GA gene of M-60(5)

The sequenced data analysis revealed the size of genomic DNA and cDNA of GA genes as 2,039 bp and 2,241 bp, respectively, with the difference of four introns varying in sizes of 49 bp, 52 bp, 45 bp and 56 bp. The open reading frame (ORF) of mutant and parent GA genes

consisted of 2,039 bp encoding 612 amino acid residues. The multiple sequence alignment of amino acid sequences from parent *A. oryzae* and mutant M-60(5) along with *A. niger* and *A. Awamori* as a reference has shown that γ-rays treatment resulted into a point mutation at nucleotide position 703, where the cytosine was replaced by adenine that resulted in a change of amino acid in catalytic site, i.e., Leu at position 203 into Isoleucine (Fig 1).

Furthermore, the 3D structure of parent and mutant M-60(5) GA enzyme showed that replacement of single amino acid has made a slight change in the microenvironment of the active site (Fig 2).

## Molecular mass determination

Similar subunit molecular mass (63.1 kDa) of GAs from Parent and M-60(5) strain was found on 10% SDS-DR-PAGE, which was further confirmed by activity staining of GA. Moreover, the accuracy was maintained by a standard curve between molecular mass and $R_f$ values of the protein ladder (Fig 3, S1 Fig ). The calculated subunit mass was nearly same as 65 kDa predicted from the deduced amino acid sequences of GA mutant M-60(5) and parent strain.

## Effect of pH & enthalpy ($\Delta H_i$) of active site residues ionization

The similar optimum pH (6.0) for GA activity was observed for both *A. oryzae* parent and mutant M-60(5), while optimum pH range with about 70% activity for mutant was 3.4–6.5, whereas for the parental enzyme it was 5.0–6.5. The ionizable groups of active side residues involved in the maximum velocity ($V_{max}$) were determined by applying Dixon plot (Fig 4). The p$Ka$ values define as ionization constant which describes the effect of pH on chemical shift or of an enzyme's activity in a reaction. The present study revealed the p$Ka_1$ of proton-donating ionizable group of parent and mutant were 4.5 and 4.55, respectively, while p$Ka_2$ of proton-receiving group for the parent was 6.5 and for mutant M-60(5) was 6.7. Carboxylic acid and imidazole were acting as ionizable groups for acidic and basic limbs, respectively in active site of both the parent and mutated Koji GA at 50 °C.

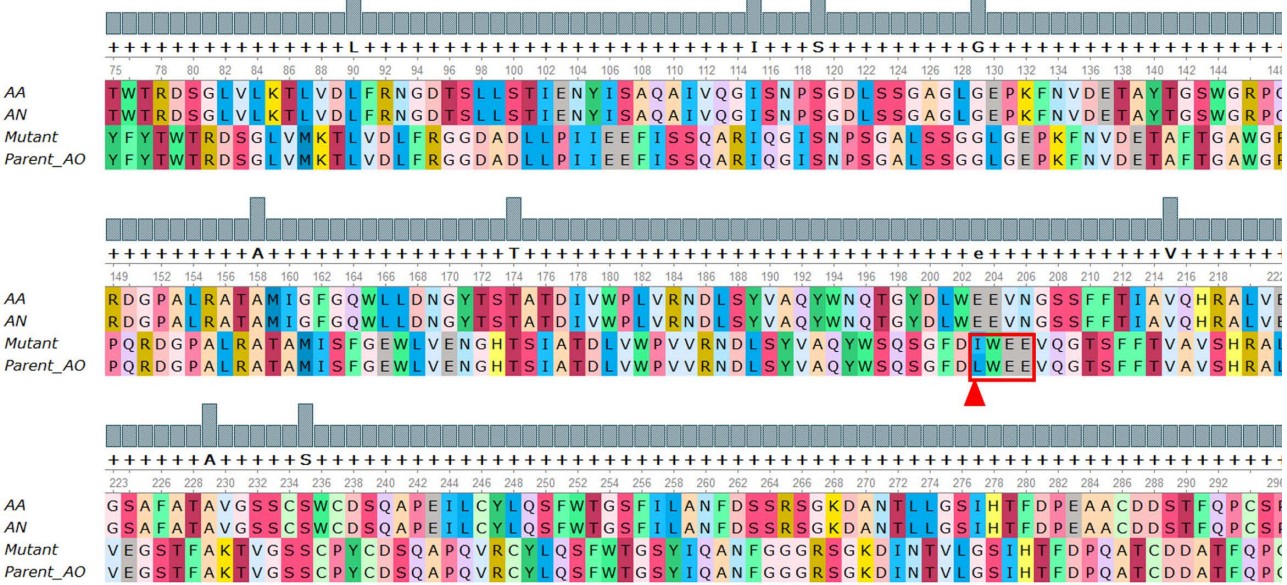

**Fig 1. Multiple sequence alignment of *A. Awamori*, *A. nigar*, *A. oryzae* parent and Mutant M-60(5) amino acid sequences.** The point mutation is highlighted in a red colored rectangle pointed by red triangle at the bottom.

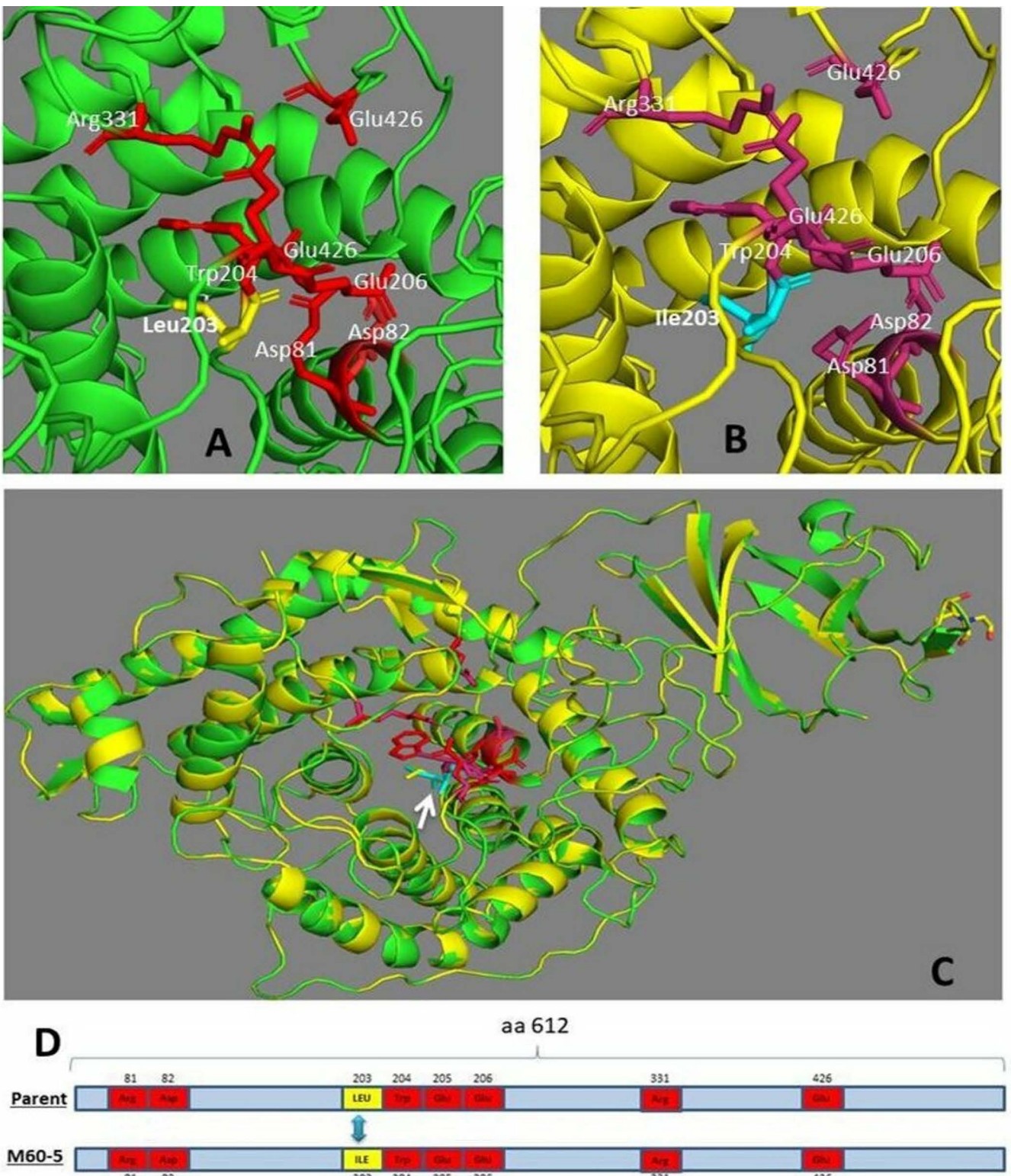

**Fig 2. Structural comparison between GAs from *A. oryzae* parent and mutant M-60(5 ).** (A) Active site of parent GA with Leucine at 203 position, (B) Active site of Mutant M-60(5) GA with Ile at 203 position and (C) superimposition of the predicted GA model of Parent (green) and the structure of GA from Mutant M-60(5) (yellow). Residues involved in substrate recognition and in catalytic site are shown in magenta and red, while the replacement of Leucine in parent into Isoleucine in mutant are shown in cyan and yellow and indicated with the arrow (white). (D) The schematic diagram to point out the mutation with yellow color.

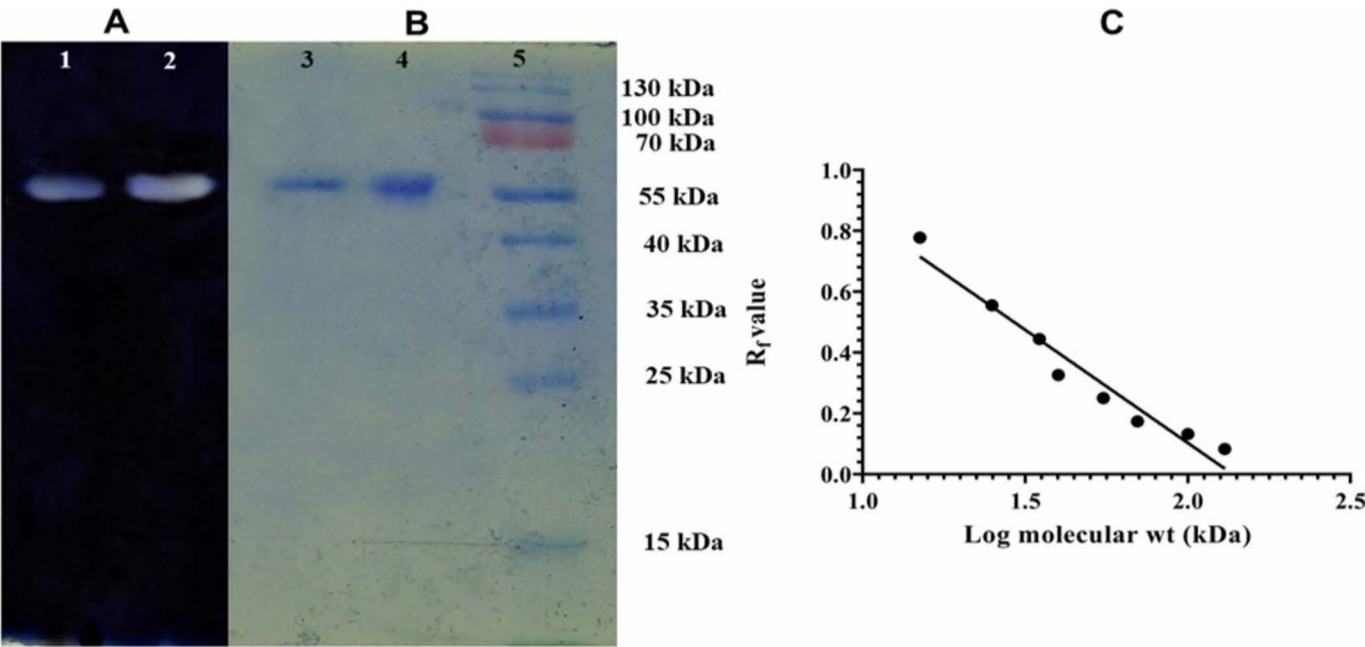

**Fig 3. Determination of subunit molecular mass of glucoamylase (GA) produced by *A. oryzae* parent and mutant M-60(5).** A): Zymographic analysis: Lane-1 Parent GA, Lane-2 M-60(5) GA. B): Lane-3 purified parent GA, Lane-4 M60(5) GA, Lane-5 protein markers [130, 100, 70(Red band), 55, 40, 35, 25 kDa] stained by Coomassie brilliant blue R-250 stain. C): Standard curve for protein markers ladder to determine the subunit molecular mass of GAs produced by *A. oryzae* strains.

Furthermore, effect of γ-rays on the conformation of GA active site was assessed indirectly by evaluating the heat of ionization ($\Delta H_i$) of ionizable groups of active site residues (Fig 5). The $\Delta H_i$ of proton donating residue for parent and mutant GA was 1054 cal mol$^{-1}$ and 1461 cal mol$^{-1}$, while for the proton receiving residue was 6576 cal mol$^{-1}$ and 6581 cal mol$^{-1}$, respectively. Therefore, we considered the conformational change in active site of mutant's GA might be due to the π MO ionization of aromatic histidine and NH$_2$ nitrogen lone pair ionization of Glu/Asp residues.

## Temperature optimum, activation energy & temperature quotient

The GA from *A. oryzae* parent and mutant M-60(5) exhibited similar optimum temp, i.e., 50 °C with optimum temp range of 45–55 °C (~80% of optimal activity). The working ability of an enzyme at elevated temperature is referred as its thermophilicity, while the resistance against unfolding at higher temperature is termed as thermostability of an enzyme. The activation energy ($E_a$) determined by Arrhenius plot for the soluble starch hydrolysis for parent GA was 50.4 kJ mol$^{-1}$, while for the mutant M-60(5) it was 46.9 kJ mol$^{-1}$ (Fig 6). Lower energy requirement by the mutant GA indicated that it was better as compared to the parent GA. Hence, we considered the mutation due to γ-rays treatment might have changed the microenvironment of the active site, resulting into improved conformation, which made the mutant M-60(5) GA more efficient in making the transition-state complex (ES*).

## Kinetics & thermodynamics of substrate hydrolysis

Effect of mutation on the Michaelis Menten kinetics constants ($K_m$, $V_{max}$, $k_{cat}$ & $k_{cat}/K_m$) of soluble starch hydrolysis by the Koji GAs were analyzed by fitting the data to non-linear

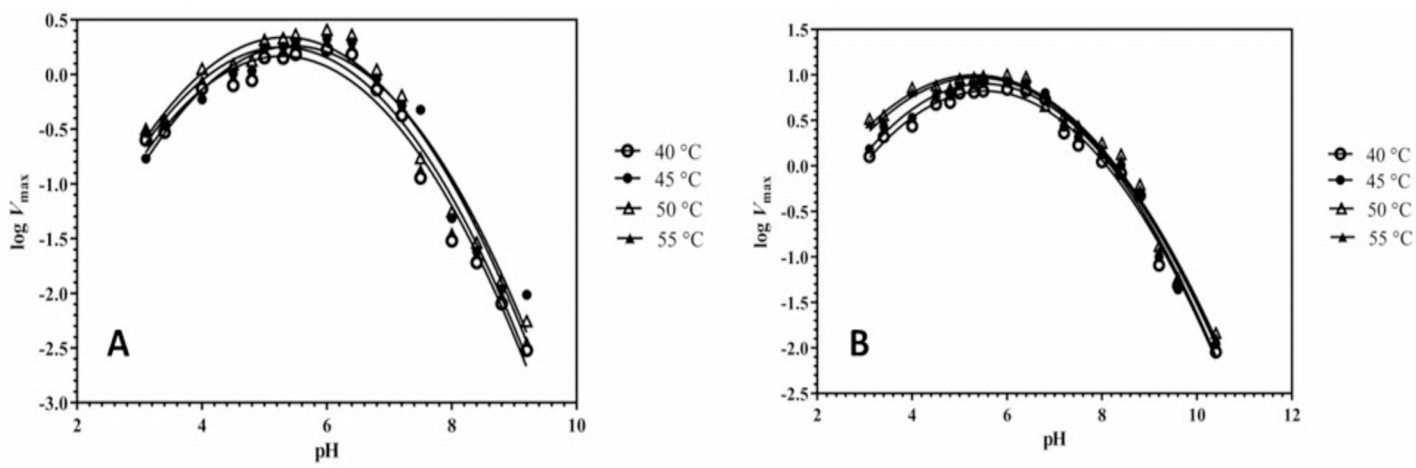

**Fig 4.  Dixon plots for the determination of p*K*a values of active site residues of GAs from (A) *A. oryzae* parent, (B) *A. oryzae* M-60(5) controlling maximum velocity for soluble starch hydrolysis at various temperatures.**

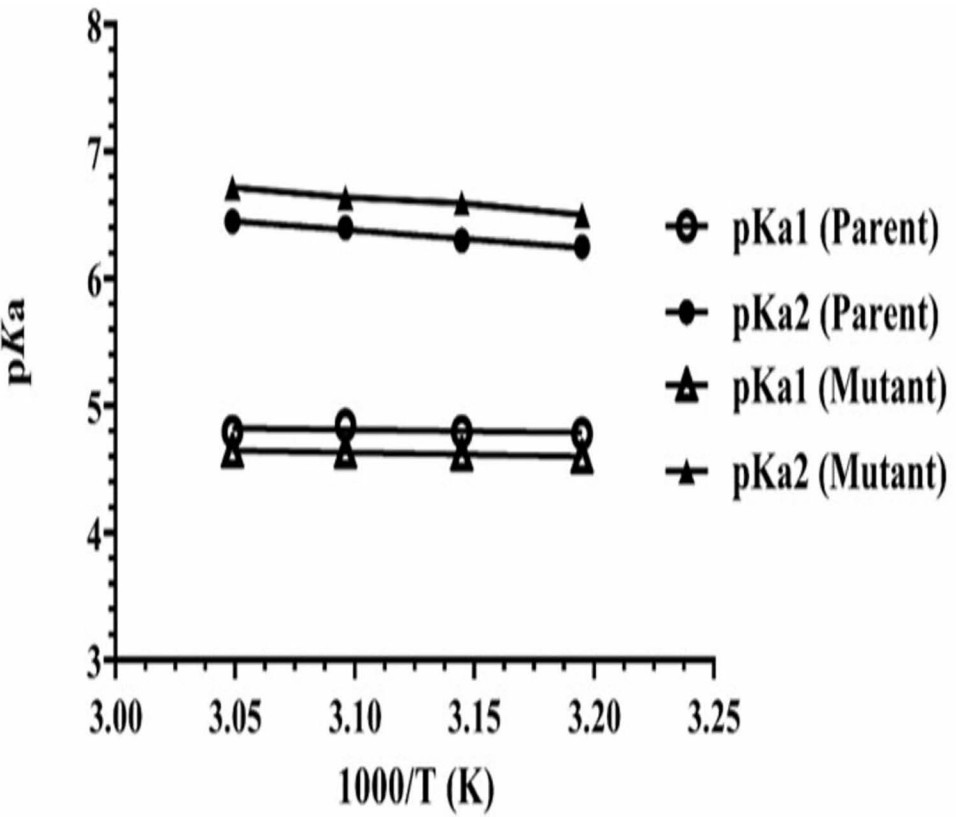

**Fig 5.   Determining heat of ionization ( $\Delta H_i$) of the active site residues of GAs from *A. oryzae* parent and mutant M-60(5) strains by Dixon plot.**

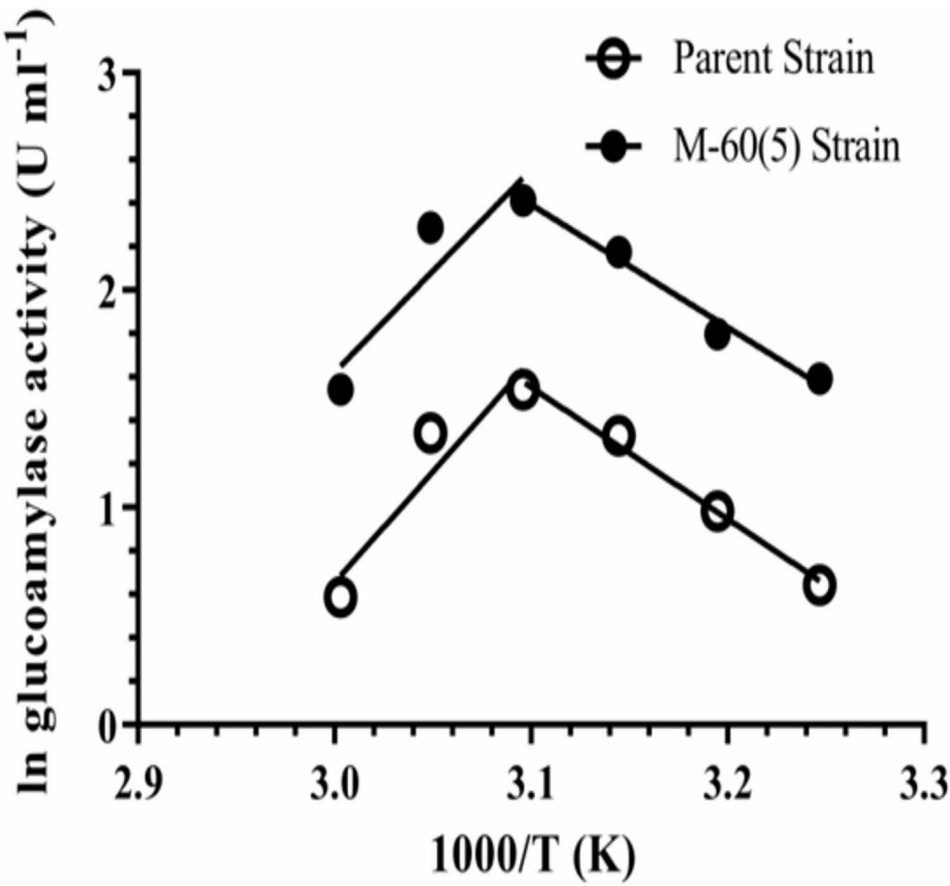

**Fig 6. Arrhenius plot to determine the activation energy for soluble starch hydrolysis by GA from *A. oryzae* parent and mutant M-60(5) strains.**

regression using the Graphpad Prism software (Fig 7). The kinetics of soluble starch hydrolysis by mutant GA was drastically improved due to replacement of Leu203 with Ile203. The $K_m$, which is the measure of binding affinity of substrate with the enzyme, of mutant M-60(5) GA for soluble starch hydrolysis at 50 °C, pH 5.0 was lower (0.06 mg ml$^{-1}$) than the parental GA having 0.17 mg ml$^{-1}$. Hence, the γ-rays induced mutation in Koji mutant M-60(5) has changed the conformation of active site and made it 2.8 folds more efficient to make the ES complex (Table 1).

The $K_{cat}$ (turn over) catalytic events performed by active site of mutated GA were 1.7 fold higher than the parent GA, i.e., 118 s$^{-1}$ (Table 1). Furthermore, substrate specificity constant ($k_{cat}/K_m$) of mutated GA was 1899 and was 4.7 fold higher than control. Since it gives information about substrate specificity when its concentration is extremely low ($<<K_m$), the results pointed towards the alternation in active site conformation. We concluded that mutation in active site residues made the mutant GA highly specific for starch hydrolysis. The catalytic efficiency of GA produce of *A. oryzae* mutant M-60(5) was extremely higher by having $K_m$ of GA ($K_m$ = 0.062 mg ml$^{-1}$).

The Gibbs free energy for soluble starch hydrolysis (Δ$G$*) of M-60(5) GA was decreased as compared to parental GA, hence, confirmed that it required lower amount of functional energy to convert the transition complex (ES*) into products. Moreover, reduction in Δ$H$* of mutated GA with 3.492 kJ mol$^{-1}$ also confirmed that it required lower energy to make the

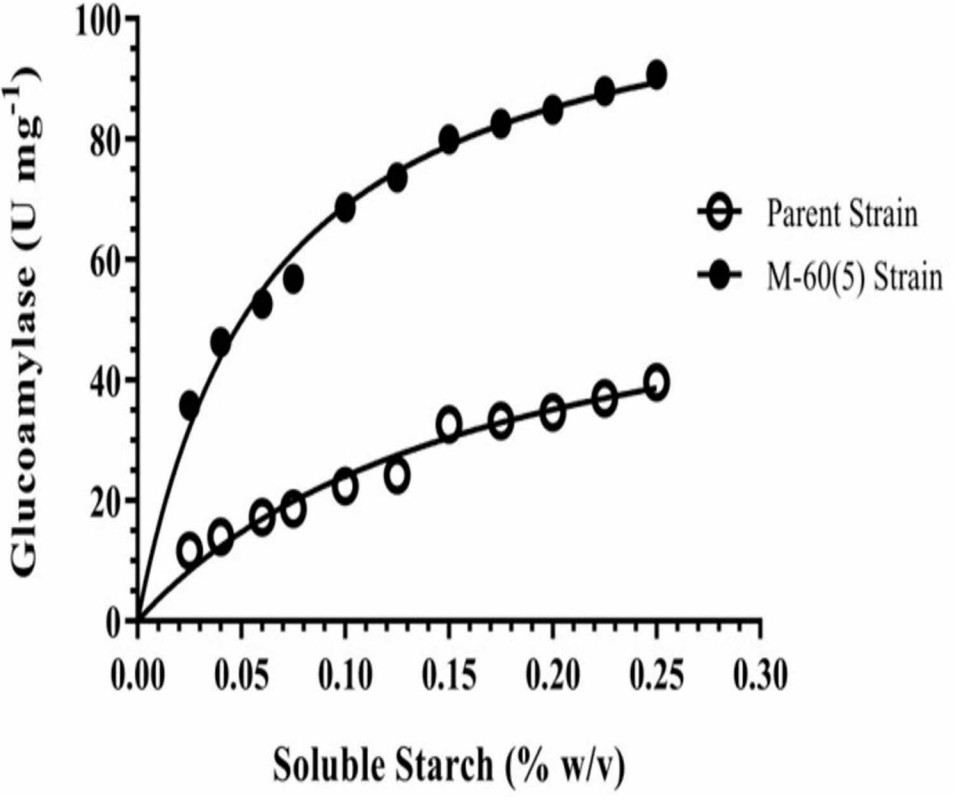

**Fig 7. Michaelis-Menten kinetic (*Vmax, Km*) constants for the soluble starch hydrolysis by GA from *A. oryzae* parent and mutant M-60(5) at 50 °C, pH 5.0.**

**Table 1. Kinetics and thermodynamics of soluble starch hydrolysis by GA produce of *A. oryzae* parent and mutant M-60(5) at 50 °C and pH 5.0.**

| Description | Parent | Mutant M-60(5) |
|---|---|---|
| $V_{max}$ (U mg$^{-1}$) | 65.01 | 112 |
| $K_m$ (mg ml$^{-1}$) | 0.171 | 0.062 |
| $V_{max}/K_m$ | 381 | 1806 |
| $K_{cat}$ (s$^{-1}$) | 68.4 | 118 |
| $K_{cat}/K_m$ | 400 | 1899 |
| $\Delta H^*$ kJ mol$^{-1}$ | 47.755 | 44.263 |
| $\Delta G^*$ kJ mol$^{-1}$ | 67.975 | 66.514 |
| $\Delta S^*$ J mol$^{-1}$ K$^{-1}$ | −62.6 | −68.9 |

activated transition complex (ES*). On the other hand, $\Delta S^*$ for starch hydrolysis of mutant GA was decreased, which confirmed that the activation in substrate hydrolysis was not entropically driven (Table 1).

## Kinetics and thermodynamics of irreversible thermostability

The thermostability of mutant GA was increased about two fold at 55 °C than the parental enzyme, while at 45 °C & 50 °C a modest increase in the stability was observed (Fig 8). The

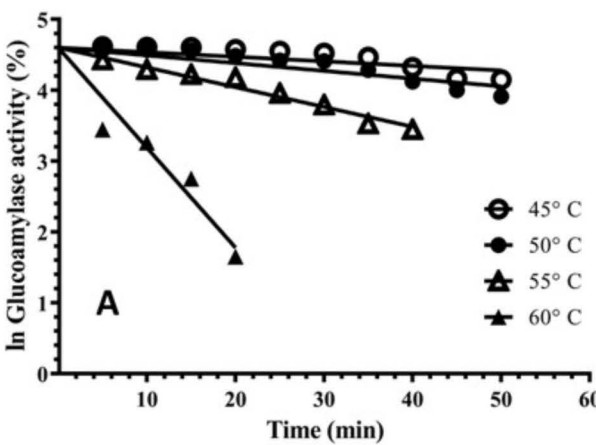 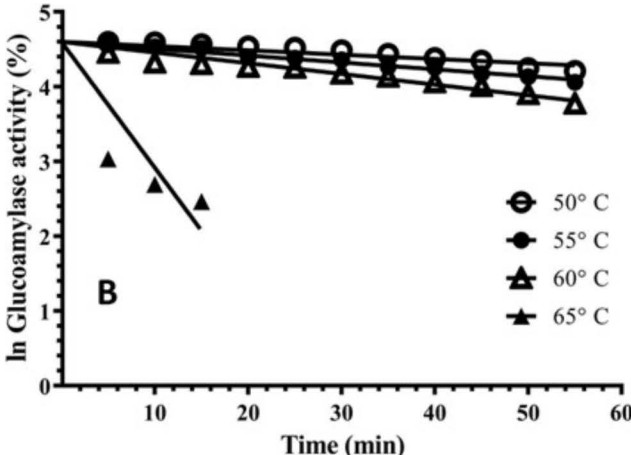

**Fig 8. This figure presents irreversible thermal inactivation of GA from (A):** *A. oryzae* **parent, (B):** *A. oryzae* **M-60(5) at variable temperatures using Pseudo first order plots.**

increase in thermal stability of M-60(5) GA was due to higher Gibbs free energy, which was increased with a difference of 3.43 kJ mol$^{-1}$. The higher free energy helped the mutated GA to resist against thermal unfolding of its transition state (U*) into an inactive enzyme.

Moreover, activation energy $E_{a(d)}$ to make the transition state 'U*' of mutant GA was also higher than the parental one, pointing towards higher stability of the mutated GA (Fig 9). The change in entropy ($\Delta S$*) of mutated GA at temperatures ranging from 45°C–60°C was higher than the parental GA, which indicated towards the increase in disorder of its active site conformation (Table 2). It was concluded that the increase in thermostability of mutant GA was due to higher $\Delta G$* and was not entropically driven.

## Toxin analysis on LC-MS

The Mutant M-60(5) GA was analyzed for toxin analysis on LC-MS against standard and already reports control values. It was confirmed that the M-60(5) GA did not produce any aflatoxins (Fig 10).

## Discussion

*A. oryzae* is used in food processing industries for centuries. Therefore, numerous attempts have been made to improve the *A. oryzae* strains for making process more feasible. Various methods are used for strain improvement, e.g., random mutagenesis [18]. Hence, in the current study, point mutation, i.e., replacement of catalytic center leucine to iso-leucine on the active site conformation was introduced in *A. oryzae* for improving catalytic efficiency and thermostability of Glucoamylase (GA) by γ-rays induced random mutagenesis.

In the current study, the GA was produced 2.6 fold higher (9.7 U ml$^{-1}$) by mutant M-60(5) as compared to parent (3.6 U ml$^{-1}$). Whereas, the specific activity of the mutant was 1.83 fold increased (54.9 U mg$^{-1}$) as compare to parent. The increase in GA enzyme activity in *A. oryzae* by γ-rays mutagenesis is confirmed in an already reported study [32]. The mutant M-60(5) strain produces much higher GAs than already reported strains, i.e., the maximum units of GA under optimum conditions are reported as 3.5 U ml$^{-1}$ from *A. wentii* [33], 8.23 U mL$^{-1}$ from *A. Oryzae* [34] and 5.9 U mg$^{-1}$ from *A. Awamori* [35]. The crude GAs from *A. oryzae* parent and Mutant M-60(5) were purified by ammonium sulphate precipitation as described

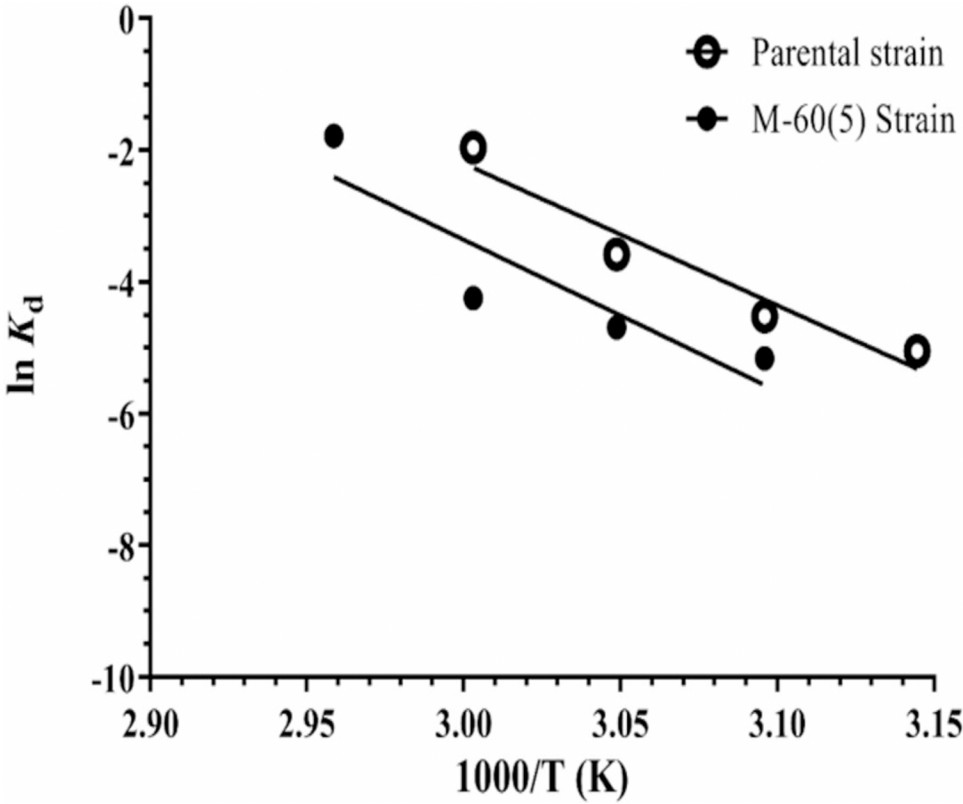

**Fig 9. Determining activation energy for irreversible thermal denaturation of GA from *A. oryzae* parent and mutant M-60(5) strains by Arrhenius plot.**

**Table 2. Kinetics and thermodynamics of irreversible thermal stability of GA from parent and mutant M-60(5) *A. oryzae*.**

| Temp. (°C) | Temp. (K) | $K_d$ (min$^{-1}$) | | t ½ (min) | | ΔH* (kJ mol$^{-1}$) | | ΔG* (kJ mol$^{-1}$) | | ΔS* (J mol$^{-1}$K$^{-1}$) | |
|---|---|---|---|---|---|---|---|---|---|---|---|
| | | P | M | P | M | P | M | P | M | P | M |
| 45 | 318 | 0.0064 | 0.0058 | 108 | 121 | 176.36 | 188.29 | 102.22 | 104.18 | 233.15 | 260.40 |
| 50 | 323 | 0.0109 | 0.0092 | 64 | 75 | 176.31 | 188.25 | 102.45 | 104.54 | 228.69 | 255.20 |
| 55 | 328 | 0.027 | 0.0143 | 25 | 48 | 176.27 | 188.20 | 101.52 | 104.95 | 227.92 | 250.01 |
| 60 | 333 | 0.141 | 0.168 | 5 | 4 | 176.23 | 188.16 | 98.62 | 99.64 | 233.08 | 261.89 |

***Keys;*** P = parent GA; M = mutant M-60(5) GA. The $Ea_{(d)}$ of parent and mutated GAs was 179.00 and 190.972 kJ mol$^{-1}$, respectively.

[26,36,]. After purification, the specific activity of the purified GAs from parent and mutant M-60(5) Koji strains was increased to 51.8 and 96.2 U mg$^{-1}$, respectively which are much higher than the specific activity of purified GA from *A. niger, i.e.,* 16.2 U mg$^{-1}$ by three steps purification [17], however the specific activity of purified GA from *A. fumigatus* was reported as 94 U mg$^{-1}$ [37].

In addition, the multiple sequence alignment of amino acid sequences from parent *A. oryzae* and mutant M-60(5) along with *A. niger* and *A. Awamori* as a reference has shown that γ-rays treatment resulted into a point mutation at nucleotide position 703, where the cytosine was replaced by adenine that resulted in a change of amino acid in catalytic site, i.e., Leu at

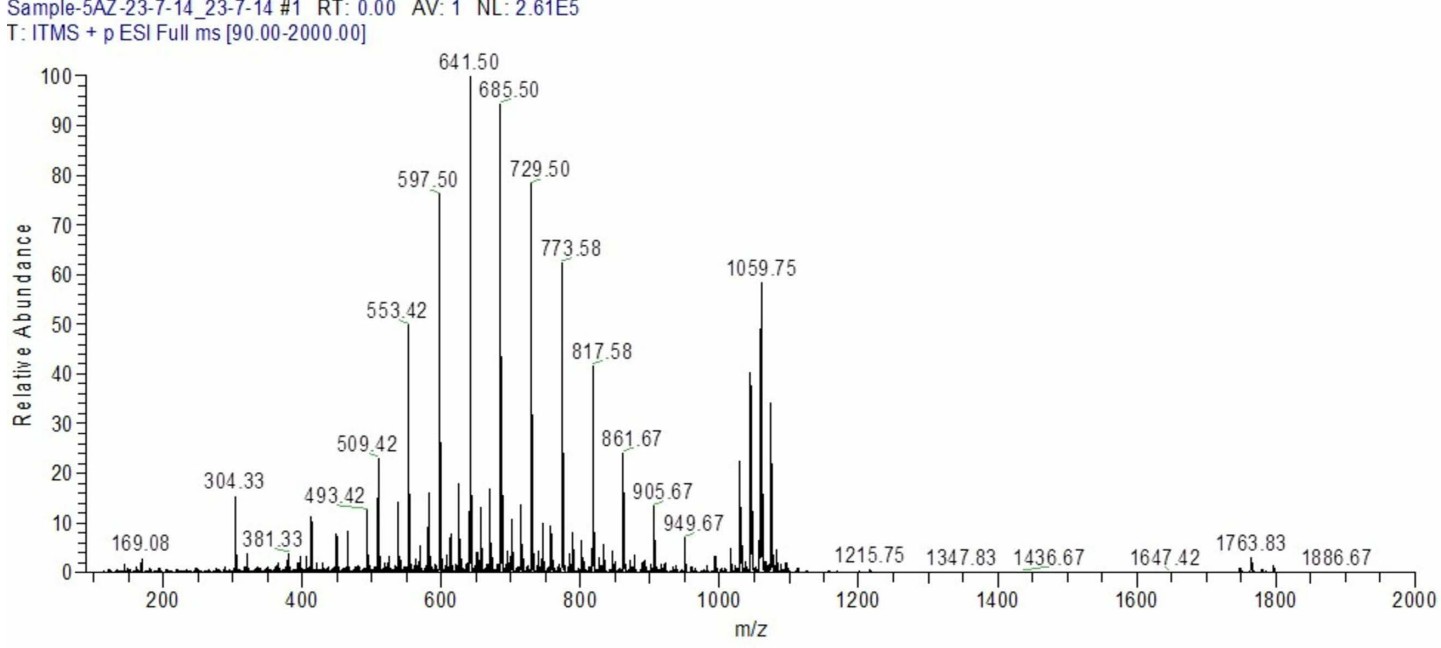

**Fig 10. Aflatoxin analysis of Mutant M- 60(5) through LC-MS.**

position 203 into Isoleucine. In previous studies, a single replacement of leucine to isoleucine in active site has drastically effect the enzyme activity of Taq polymerase by converting it into highly cold sensitive enzyme [38], whereas as a result of mutagenesis, a single conserved amino acid replacement, i.e., leucine to isoleucine has affected the 'g' protein signals mechanism [39]. Hence, based on the above report we concluded that the replacement of leucine203 into isoleucine in M-60(5) Koji GA might have drastically changed the conformation of its active site, resulting into an efficient and stable mutant GA enzyme. We believe that improvement in catalytic efficiency of GAs in mutant was linked with the change in a Leu203 to Ile203 at active site of GA Mutant M-60(5).

In the present study, the molecular mass of GAs was determined and fund subunit molecular mass (63.1 kDa) of GAs from Parent and M-60(5). Moreover, the accuracy was maintained by a standard curve between molecular mass and $R_f$ values of the protein ladder. Mutant M-60(5) showed almost similar molecular mass as of GA from *Aspergillus oryzae* from Luzhou-flavour Daqu [40] and *A. Fumigatus* [37]. Already reported studies supports our finding of similar subunit molecular mass of parent and mutant strains from *Aspergillus oryzae* [41] and from *A. flavus* [42]. These findings are in accordance with other published results, where the GA from *A. flavonus* shown protein size of 78 kDa which varied from the calculated size of 55.1 kDa [34].

Furthermore, optimum pH range with showing 70% activity for mutant was between 3.4–6.5. The more acidic pH range of mutant GA indicated that the γ-rays mediated point mutation has slightly changed the microenvironment of the active site of mutant GA. Previous studies showed the fungal GA remains more active at acidic pH [43,44]. The maximum catalytic activity of this enzyme has also been reported at pH 5 to 6, whereas in *A. Niger* the optimum reported as pH 4.8 [45]. The present study revealed the p$Ka_1$ of proton-donating ionizable group of parent and mutant were 4.5 and 4.55, respectively, while p$Ka_2$ of proton-receiving group for the parent was 6.5 and for mutant M-60(5) was 6.7. The difference in p$Ka$

values of acidic and basic limbs of the active site residues of mutated GA confirmed about the changed configuration of the active site due to random mutagenesis resulting into more efficient catalytic activity of mutant M-60(5). Literature confirms that the change in microenvironment around the catalytic domain resulting into improvement in catalytic efficiency and protein stability at low pH [46]. The reported ionizable groups values of amino acids located in proteins indicated the presence of glutamic/aspartic acid as the proton donating residue, while histidine as the proton receiver [29].

The $\Delta H_i$ of proton donating residue for parent and mutant GA was 1054 cal mol$^{-1}$ and 1461 cal mol$^{-1}$, while for the proton receiving residue was 6576 cal mol$^{-1}$ and 6581 cal mol$^{-1}$, respectively. Similar findings regarding proton donating residues were presented by [15], where three Glutamic acid (Glu) and one Aspartic acid (Asp) residues participated as the electron donor in GA active site. Effect of temp on p$Ka_1$ & p$Ka_2$ and enthalpy or heat of ionization ($\Delta H_i$) of active site residues ionizable groups was determined as described by Dixon and Web [29] (Fig 5). The amino acid composition of mutant's active site was not changed, however, the increase in $\Delta H_i$ of proton donating (407 cal mol$^{-1}$) and receiving (5 cal mol$^{-1}$) residues evidenced about the change in conformation of active site. Dehareng and Dive [47] evaluated the ionization energies (IE) as a function for the conformation of α-L-amino acids and optimized three to five conformations for the arginine, lysine, isoleucine, tyrosine and tryptophan in their study.

Similarly, the GA from *A. oryzae* parent and mutant M-60(5) exhibited similar optimum temp, i.e., 50 °C with optimum temp range of 45–55 °C (~80% of optimal activity), which is considered as much higher optimal activity at given range than already reported *A. flavus*, which showed a decline in optimal activity above 50 °C, pH 5.5 [42]. The optimum temp of GA for soluble starch was reported as 55 °C from *Gymnoascella citrina* [44], whereas, the purified intracellular GA from *A. tritic i* WZ99 gave temp optimum of 45 °C [48].

Michaelis Menten kinetics constants ($K_m$, $V_{max}$, $k_{cat}$ & $k_{cat}/K_m$) of soluble starch hydrolysis by the mutant and parent Koji GAs were determined in the current study. The $K_m$ of mutant M-60(5) GA for soluble starch hydrolysis at 50 °C and pH 5.0 was lower (0.06 mg ml$^{-1}$) than the parental GA having 0.17 mg ml$^{-1}$. Previous studies reported lesser km values of naïve and modified enzyme, i.e., 0.34 and 0.29, respectively [49] and 2.1mg/ml of glucoamylase GA-LZ2 than the km values reported in this study [40]. The $K_{cat}$ (turn over) catalytic events performed by active site of mutated GA were 1.7 fold higher than the parent GA, i.e., 118 s$^{-1}$. Moreover, turnover of the mutant GA was higher than that of recently reported novel mesophilic GA having 67.15 s$^{-1}$ [48] and GA of *A. oryzae* with 20.3s$^{-1}$ from Luzhou-flavour Daqu [40]. We believed that the mutation has altered the conformation of active site of mutant GA and made it more flexible and efficient to convert the transition ES*-complex into products. Furthermore, substrate specificity constant ($k_{cat}/K_m$) of mutated GA was 1899 and was 4.7 fold higher than control. Similar findings are reported in literature that γ-rays based mutagenesis of *A. niger* significantly improved the kinetic properties of GAs from mutant strains for starch hydrolysis [17]. Similar trend in kinetic parameters due to the γ-rays mediated mutagenesis in *A. niger* for lignocellulose hydrolysis by β-glycosidase was reported by Javed et al [58]. In another report Karim et al. [34] reported $K_m$ and $V_{max}$ for soluble starch as 5.84 mg ml$^{-1}$ and 153.85 U mg$^{-1}$ respectively for GAs from *A. flavus* NSH9. The current report indicates that mutant M-60(5) is highly substrate specific as compare to the already reported studies.

In addition, the catalytic efficiency of GA produce of *A. oryzae* mutant M-60(5) was extremely higher than the salt tolerant *A. flavus*, which had $K_m$ of 0.72 mg ml$^{-1}$ with $V_{max}$ of 12.48 µ mol min$^{-1}$ mg$^{-1}$. The $K_m$ of mutated GA ($K_m$ = 0.062 mg ml$^{-1}$) highlighted that its affinity to soluble starch was about 11.6 fold higher, while maximum velocity ($V_{max}$ = 112 µmol min$^{-1}$ mg$^{-1}$) was about 9.0 fold higher than that of *A. flavus* GA [42]. Hence, in comparison to previous studies, the kinetic properties of GA from mutant M-60(5) exhibited excellent

catalytic activity in terms of starch hydrolysis that make efficient use of enzyme in industry. Similarly, we already reported γ-rays mutagenesis-based improvement in kinetic properties on thermodynamics of starch hydrolysis, where $\Delta H^*$ (kJ mol$^{-1}$), $\Delta G^*$ (kJ mol$^{-1}$), $\Delta S^*$ (J mol$^{-1}$K$^{-1}$) for parent and mutant were 41.50, 46.12; 65.69, 63.62; −72.65, −52.53, respectively [17]. Hence, we concluded that the mutated GA was thermodynamically more efficient in conversion of soluble starch into products.

In the current study, the thermostability of mutant GA was increased about two fold at 55 °C than the parental enzyme, while at 45 °C & 50 °C a modest increase in the stability was observed (Fig 8 A, B). The GA from *A. brasiliensis* showed half-life of 22 min at 55 °C [50], while half-life of mutant GA of *A. awamori* at 55 °C was 48 min [51], which is less than that of mutant GA M-60(5). Previously, it was reported that γ-rays mutagenesis has affected *A. niger* in terms of thermodynamic parameters for cellobiose hydrolysis and improved the thermal stability of mutant enzyme. Where energy of activation ($Ea_{(d)}$) of parent strain was 274 kJ mol$^{-1}$, while for mutant strain it was 240 kJ mol$^{-1}$. The $\Delta H^*$ reported for mutant was as 237.02 kJ mol$^{-1}$, while for parent was 271.44 kJ mol$^{-1}$, whereas $\Delta G^*$ was 105.67 kJ mol$^{-1}$ for mutant and 105.49 kJ mol$^{-1}$ for parent. The change in entropy ($\Delta S^*$) for mutant was 407.90 J mol$^{-1}$K$^{-1}$, while for parent 515.35 J mol$^{-1}$K$^{-1}$ [52].

## Conclusions

Random mutagenesis of *A. oryzae* by γ-ray treatment simultaneously enhanced the productivity, catalytic efficiency and thermostability of GA. The improvement in stability and function of GA was due to a point mutation in active site of GA encoding gene, resulted in replacement of amino acid Leu to Ile at position 203. Hence, the mutation changed the microenvironment of GA active site, resulting into alteration in active site conformation of mutant M-60(5) GA. The changed p$K$a values, $\Delta H_i$ of acidic & basic limbs of the active site residues of mutated GA evidenced about the alteration of active site conformation. We concluded conformational change in active site of mutant GA was due to π MO ionization of aromatic histidine and NH$_2$ nitrogen lone pair ionization of Glu/Asp residues. Thermostabilization of mutant GA was due to higher $\Delta G^*$. We concluded stability-function of enzymes might be simultaneously enhanced by strain improvement through γ-rays treatment. The mutated GA due to its high catalytic efficiency and thermostability proved that it has great potential for application in food industry such as beverages, baking and starch saccharification.

## Supporting information

**S1 File. Raw images.**
(PDF)

## Author agreement

The final version of the submitted manuscript has read and approved by all authors. It is hereby confirmed that this manuscript is in original form and not submitted for publication elsewhere.

## Acknowledgments

We are thankful to Mr. Zubair Nawaz Chattha, the Chief Executive of Gourmet Pvt. Limited, Pakistan and Ms. Fatima Nawaz, the Director of QuinTech Center for Applied Sciences (QCAS), Lahore for providing technical support for the research work. Also, Mr. Ghulam Ali Waseer (late) is gratefully acknowledged for providing his assistance toward the conducted research.

## Author contributions

**Conceptualization:** Anam Saqib.

**Data curation:** Anam Saqib, Saif -ur-Rehman, Hazrat Ali, Asad Ali.

**Formal analysis:** Anam Saqib, Asad Ali.

**Methodology:** Anam Saqib, Asad Ali.

**Software:** Noor Hassan.

**Supervision:** Hazrat Ali, Muhammad Hamid Rashid.

**Validation:** Noor Hassan, Muhammad Hamid Rashid.

**Visualization:** Saif -ur-Rehman, Hazrat Ali.

**Writing – original draft:** Anam Saqib.

**Writing – review & editing:** Hazrat Ali, Noor Hassan, Muhammad Hamid Rashid.

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
