## [Decision Letter · Decision Letter 0]

16 Dec 2024

PONE-D-24-53783Gama Rays Mediated Improvement of Catalytic Efficiency and Thermostability of Glucoamylase by Replacing Active Site Leucine to Isoleucene from Super Koji (Aspergillus oryzae)PLOS ONE

Dear Dr. Ali,

Thank you for submitting your manuscript to PLOS ONE. After careful consideration, we feel that it has merit but does not fully meet PLOS ONE’s publication criteria as it currently stands. Therefore, we invite you to submit a revised version of the manuscript that addresses the points raised during the review process.

We look forward to receiving your revised manuscript.

Kind regards,

Habibullah Nadeem

Academic Editor

PLOS ONE

Journal Requirements:

3. We note that your Data Availability Statement is currently as follows: “All relevant data are within the manuscript and in Supporting Information files.”

Reviewers' comments:

Reviewer's Responses to Questions

**Comments to the Author**

1. Is the manuscript technically sound, and do the data support the conclusions?

Reviewer #1: Yes

Reviewer #2: Yes

2. Has the statistical analysis been performed appropriately and rigorously? 

Reviewer #1: N/A

Reviewer #2: Yes

3. Have the authors made all data underlying the findings in their manuscript fully available?

Reviewer #1: Yes

Reviewer #2: Yes

4. Is the manuscript presented in an intelligible fashion and written in standard English?

Reviewer #1: Yes

Reviewer #2: Yes

5. Review Comments to the Author

Reviewer #1: The manuscript entitled "Gama Rays Mediated Improvement of Catalytic Efficiency and Thermostability of Glucoamylase by Replacing Active Site Leucine to Isoleucine from Super Koji (Aspergillus oryzae)” presents results of (random) gamma-ray induced mutation of the A. oryzae glucoamylase, which results in one positive point mutation on the enzyme that shows that the single point mutation L103I provides improved thermostability of the enzyme and improved kcat. The manuscript is well fitted to the journal scope.

So, I recommend that the manuscript to be accepted for publication in PLOS ONE. The data could be of high interest for scientific community working in the field. However, before publication the author should addressed the following minor changes / points;

INTRODUCTION

The introduction part clearly summarizes the current state of the topic and addresses the limitations of current knowledge in this field with clear and appropriate research question. However, the following minor changes need to addressed by authors;

Line 44: Comma missing after ‘However’.

Line 44-45: Replace the sentence ‘However their major short coming of low catalytic activity lead to consume high energy and restrictions in starch processing and fermentation’ with ‘However, their lower catalytic activity leads to consume high energy and restrictions in starch processing and fermentation’.

Line 61-63: Add reference (s) for the sentence ‘Random mutagenesis induced by γ-rays may contribute in enhancement of industrial enzymes production; hence γ-ray mediated mutagenesis of A. niger made the mutant GA highly efficient in substrate hydrolysis’.

Line 65: Comma missing after ‘Previously’.

Line 71: Remove the ‘data not shown’.

MATERIALS AND METHODS

The study design and methods are appropriate for the research question with enough detail for each experiment. However, the following minor changes need to addressed by authors;

Line 110: Add the paragraph under ‘In silico sequence analysis’ to heading ‘Sequencing and in-silico point mutation identification in GA gene’.

Line 113-112: Add full name for first time to ‘EMBOSS, EMBL-EBI and NCBI’.

Line 139: Remove ‘s’ from GA.

RESULTS

The authors presented results clearly and accurately and all the relevant data been included as well as the data described in the text consistent with the data in the figures and tables. However, the following minor changes need to addressed by authors;

Line 205-206: Replace the sentence ‘DNA Extraction and PCR amplification of A. oryzae Mutant M-60(5) and parent strain indicated the size of genomic DNA and cDNA of GA genes as 2,039 bp and 2,241 bp, respectively’ with ‘The sequenced data analysis revealed the size of genomic DNA and cDNA of GA genes as 2,039 bp and 2,241 bp, respectively’.

Line 213: Incorrect abbreviations for A. Awamori (AA), A. nigar (AN), A. oryzae (AO). Write the correct abbreviations.

Line 255: Replace the word ‘consider’ with ‘considered’.

Line 267: Replace the word ‘consider’ with ‘considered’.

DISCUSSION

The authors logically explained the findings and compared the findings with current findings in the research field. However, the following minor changes need to addressed by authors;

Line 351: Replace the word ‘may’ with ‘might’.

Line 352-354: Replace the sentence ‘In the current study, it is supposed that the improvement in catalytic efficiency of GAs in mutant was linked with the change in a Leu203 to Ile203 at active site of GA Mutant M-60(5)’ with ‘we believe that improvement in catalytic efficiency of GAs in mutant was linked with the change in a Leu203 to Ile203 at active site of GA Mutant M-60(5)’.

Line 391-392: The sentence ‘that shows decline in optimal activity above 50 °C, pH 5.5’ should be written as ‘which showed a decline in optimal activity above 50 °C, pH 5.5’.

Line 419-421: Replace the sentence ‘Similarly, previously we had reported about the improvement in kinetic properties due to γ-rays mutagenesis on thermodynamics of starch hydrolysis’ with ‘Similarly, we already reported γ-rays mutagenesis-based improvement in kinetic properties on thermodynamics of starch hydrolysis’.

CONCLUSION

The authors logically concluded the whole research work but needs to be addressed the following minor changes.

Line 441: Remove the word ‘drastic’.

Line 446: Replace the word ‘may’ with ‘might’.

REFERENCES

The references are correct and missing no key reference, as per my evaluation.

Reviewer #2: The authors mutated Aspergillus oryzae, also known as Super Koji, by gama rays for enhancing catalytic efficiency and thermostability of glucoamylase by replacing active site leucine to isoleucene. The point mutation altered the conformation of the glucoamylase active site that contributed to improve the functional energy, resulted the stabilization of transition state which made it thermostable and highly efficient in starch hydrolysis. I would like to recommend the research work to be published in PLOS ONE. However, the following minor suggestions should be addressed by authors before publication.

L 43. Use full name for the first time e.g. ‘R. delemar’.

L 47. Replace word ‘their’ with ‘its’.

L 49. The sentence ‘In recent era, strain improvement by site directed and random mutagenesis of fungi is an interesting research topic’ should be replaced with ‘In recent era, strain improvement by site directed and random mutagenesis of fungi is being used widely for industrial applications’.

L 59. Use short name for ‘Aspergillus oryzae’. Same consistency must be kept throughout the manuscript.

L 65-67. Support your claim by providing reference (if published) for ‘Previously we developed novel Aspergillus oryzae mutants by γ-rays’ mutagenesis for the enhanced production of thermostable α-amylases, which were also highly specific in the α-amylase production’.

L 72. Change the sentence ‘Novelty of current report is as it for the first time explains about effect of point mutation’ with ‘The current study reports for the first time effect of point mutation’.

L 78. The word ‘Maintenance’ should be started with small letter.

L 95. Ether use ‘hours’ or ‘hrs’. Correct the same in entire manuscript.

L 98. Mention the pore size of muslin cloth.

L 99. Which instrument was used for lyophilization? Provide model and manufacturer name etc.

L 135. Add brief procedure detail to Protein assay.

L 211. Add comma after ‘nucleotide position 703’.

L 289. Correct the ‘pointed out’ to ‘pointed’.

L 289. Correct the ‘does not’ to ‘did not’.

L 344. Add comma after ‘nucleotide position 703’.

L 360. Correct the ‘Similar’ to ‘similar’.

L 360. The word ‘Parent’ should be started with small letter.

L 380. Add comma after ‘Similar findings regarding proton donating residues were presented by [15]’.

L 386-388. This sentence is ambiguous and needs to be re-phrased ‘Dehareng and Dive [47] evaluated the ionization energies (IE) as a function of the conformation for α-L-amino acids and three to five conformations for the arginine, lysine, isoleucine, tyrosine and tryptophan were optimized’.

L 390. Add comma after ‘with optimum temp range of 45–55 °C (~80% of optimal activity)’.

L 396-400. This sentence is ambiguous, long and needs to be re-phrased ‘The Km of mutant M-60(5) GA for soluble starch hydrolysis at 50 °C, pH 5.0 was lower (0.06 mg ml-1) than the parental GA having 0.17 mg ml-1. Which is highly efficient than the already reported km values of naïve and modified enzyme i.e 0.34 and 0.29 respectively and 2.1mg/ml of glucoamylase GA-LZ2’.

L 431. Add comma after ‘strain was 274 kJ mol-1’.

L 432. Add comma after ‘mutant was as 237.02 kJ mol-1’.

L 434. Add comma after ‘mutant was 407.90 J mol-1K-1’.

6. PLOS authors have the option to publish the peer review history of their article (what does this mean? ). If published, this will include your full peer review and any attached files.

**Do you want your identity to be public for this peer review?** For information about this choice, including consent withdrawal, please see our Privacy Policy .

Reviewer #1: No

Reviewer #2: No

---

## [Author Response · Author response to Decision Letter 1]

8 Jan 2025

A revised version of subject-stated manuscript (Research Article) is submitted into your esteemed journal “PLOS ONE”. All the comments from reviewers are carefully addressed and submitted to journal.

Response to Reviewers

S. No Reviewer (s) Corr. Author Reply

Reviewer # 1

INTRODUCTION

The introduction part clearly summarizes the current state of the topic and addresses the limitations of current knowledge in this field with clear and appropriate research question. However, the following minor changes need to addressed by authors;

1 Line 44: Comma missing after ‘However’. Comma is added after ‘However’. L 44

2 Line 44-45: Replace the sentence ‘However their major short coming of low catalytic activity lead to consume high energy and restrictions in starch processing and fermentation’ with ‘However, their lower catalytic activity leads to consume high energy and restrictions in starch processing and fermentation’. The sentence is being replaced. L 44-45

3 Line 61-63: Add reference (s) for the sentence ‘Random mutagenesis induced by γ-rays may contribute in enhancement of industrial enzymes production; hence γ-ray mediated mutagenesis of A. niger made the mutant GA highly efficient in substrate hydrolysis’. Reference is added. L 66

Line 65: Comma missing after ‘Previously’. Comma is added after ‘Previously’. L 68

Line 71: Remove the ‘data not shown’ The ‘data not shown’ is being removed. L 74

MATERIALS AND METHODS

The study design and methods are appropriate for the research question with enough detail for each experiment. However, the following minor changes need to addressed by authors;

Line 110: Add the paragraph under ‘In silico sequence analysis’ to heading ‘Sequencing and in-silico point mutation identification in GA gene’. Paragraph is being added under heading ‘Sequencing and in-silico point mutation identification in GA gene’. L 133-140

Line 113-112: Add full name for first time to ‘EMBOSS, EMBL-EBI and NCBI’. Full names for ‘EMBOSS, EMBL-EBI and NCBI’are added. 134-137

Line 139: Remove ‘s’ from GA. The ‘s’ is being removed. L 153

RESULTS

The authors presented results clearly and accurately and all the relevant data been included as well as the data described in the text consistent with the data in the figures and tables. However, the following minor changes need to addressed by authors;

Line 205-206: Replace the sentence ‘DNA Extraction and PCR amplification of A. oryzae Mutant M-60(5) and parent strain indicated the size of genomic DNA and cDNA of GA genes as 2,039 bp and 2,241 bp, respectively’ with ‘The sequenced data analysis revealed the size of genomic DNA and cDNA of GA genes as 2,039 bp and 2,241 bp, respectively’. The sentence is being replaced. L 219-222

Line 213: Incorrect abbreviations for A. Awamori (AA), A. nigar (AN), A. oryzae (AO). Write the correct abbreviations. The abbreviations are being corrected. L 219-230

Line 255: Replace the word ‘consider’ with ‘considered’. The word ‘consider’ is being replaced with ‘considered’. L 271.

Line 267: Replace the word ‘consider’ with ‘considered’. The word ‘consider’ is being replaced with ‘considered’. L 283.

DISCUSSION

The authors logically explained the findings and compared the findings with current findings in the research field. However, the following minor changes need to addressed by authors;

Line 351: Replace the word ‘may’ with ‘might’. The word ‘may’ is being replaced with ‘might’. L 367.

Line 352-354: Replace the sentence ‘In the current study, it is supposed that the improvement in catalytic efficiency of GAs in mutant was linked with the change in a Leu203 to Ile203 at active site of GA Mutant M-60(5)’ with ‘we believe that improvement in catalytic efficiency of GAs in mutant was linked with the change in a Leu203 to Ile203 at active site of GA Mutant M-60(5)’. The sentence is being replaced. L 368-371.

Line 391-392: The sentence ‘that shows decline in optimal activity above 50 °C, pH 5.5’ should be written as ‘which showed a decline in optimal activity above 50 °C, pH 5.5’. The sentence is being replaced. L 408-410.

Line 419-421: Replace the sentence ‘Similarly, previously we had reported about the improvement in kinetic properties due to γ-rays mutagenesis on thermodynamics of starch hydrolysis’ with ‘Similarly, we already reported γ-rays mutagenesis-based improvement in kinetic properties on thermodynamics of starch hydrolysis’. The sentence is being replaced. L 440-437.

CONCLUSION

The authors logically concluded the whole research work but needs to be addressed the following minor changes.

Line 441: Remove the word ‘drastic’. The ‘drastic’ is being removed. L 461

Line 446: Replace the word ‘may’ with ‘might’. The word ‘may’ is being replaced with ‘might’. L 47.

Reviewer # 2

1 L 43. Use full name for the first time e.g. ‘R. delemar’. Full name is added. L 43

2 L 47. Replace word ‘their’ with ‘its’. The word ‘their’ is being replaced with ‘its’. L 47.

3 L 49. The sentence ‘In recent era, strain improvement by site directed and random mutagenesis of fungi is an interesting research topic’ should be replaced with ‘In recent era, strain improvement by site directed and random mutagenesis of fungi is being used widely for industrial applications’. The sentence is being replaced. L 50-53.

4 L 59. Use short name for ‘Aspergillus oryzae’. Same consistency must be kept throughout the manuscript. Correction is being made. L 62

5 L 65-67. Support your claim by providing reference (if published) for ‘Previously we developed novel Aspergillus oryzae mutants by γ-rays’ mutagenesis for the enhanced production of thermostable α-amylases, which were also highly specific in the α-amylase production’. Reference is being provided. L 70

6 L 72. Change the sentence ‘Novelty of current report is as it for the first time explains about effect of point mutation’ with ‘The current study reports for the first time effect of point mutation’. The sentence is being changed. L 75-76

7 L 78. The word ‘Maintenance’ should be started with small letter. Correction is being made. L 82

8 L 95. Ether use ‘hours’ or ‘hrs’. Correct the same in entire manuscript. Corrections are being made. L 99, 100

9 L 98. Mention the pore size of muslin cloth. The pore size is mentioned. L 102

10 L 99. Which instrument was used for lyophilization? Provide model and manufacturer name etc. The model and manufacturer name are provided. L 103-104

L 211. Add comma after ‘nucleotide position 703’. Comma is added. L 226

L 289. Correct the ‘pointed out’ to ‘pointed’. Correction is being made. L 305

L 289. Correct the ‘does not’ to ‘did not’. Correction is being made. L 336

L 344. Add comma after ‘nucleotide position 703’. Comma is added. L 360

L 360. Correct the ‘Similar’ to ‘similar’. Correction is being made. L 376

L 360. The word ‘Parent’ should be started with small letter. Correction is being made. L 377

L 380. Add comma after ‘Similar findings regarding proton donating residues were presented by [15]’. Comma is added. L 397

L 386-388. This sentence is ambiguous and needs to be re-phrased ‘Dehareng and Dive [47] evaluated the ionization energies (IE) as a function of the conformation for α-L-amino acids and three to five conformations for the arginine, lysine, isoleucine, tyrosine and tryptophan were optimized’. The sentence is being rephrased and written clear for making sense. L 403-405

L 390. Add comma after ‘with optimum temp range of 45–55 °C (~80% of optimal activity)’. Comma is added. L 407

L 396-400. This sentence is ambiguous, long and needs to be re-phrased ‘The Km of mutant M-60(5) GA for soluble starch hydrolysis at 50 °C, pH 5.0 was lower (0.06 mg ml-1) than the parental GA having 0.17 mg ml-1. Which is highly efficient than the already reported km values of naïve and modified enzyme i.e. 0.34 and 0.29 respectively and 2.1mg/ml of glucoamylase GA-LZ2’. The sentence is being rephrased and written clear for making sense. L 414-418

L 431. Add comma after ‘strain was 274 kJ mol-1’. Comma is added. L 451

L 432. Add comma after ‘mutant was as 237.02 kJ mol-1’. Comma is added. L 452

L 434. Add comma after ‘mutant was 407.90 J mol-1K-1’. Comma is added. L 454

---

## [Decision Letter · Decision Letter 1]

30 Jan 2025

Gama Rays Mediated Improvement of Catalytic Efficiency and Thermostability of Glucoamylase by Replacing Active Site Leucine to Isoleucene from Super Koji (Aspergillus oryzae)

PONE-D-24-53783R1

Dear Dr. Ali,

We’re pleased to inform you that your manuscript has been judged scientifically suitable for publication and will be formally accepted for publication once it meets all outstanding technical requirements.

Kind regards,

Habibullah Nadeem

Academic Editor

PLOS ONE

Additional Editor Comments (optional):

Reviewers' comments:

Reviewer's Responses to Questions

**Comments to the Author**

1. If the authors have adequately addressed your comments raised in a previous round of review and you feel that this manuscript is now acceptable for publication, you may indicate that here to bypass the “Comments to the Author” section, enter your conflict of interest statement in the “Confidential to Editor” section, and submit your "Accept" recommendation.

Reviewer #1: All comments have been addressed

Reviewer #2: All comments have been addressed

2. Is the manuscript technically sound, and do the data support the conclusions?

Reviewer #1: Yes

Reviewer #2: Yes

3. Has the statistical analysis been performed appropriately and rigorously? 

Reviewer #1: I Don't Know

Reviewer #2: Yes

4. Have the authors made all data underlying the findings in their manuscript fully available?

Reviewer #1: Yes

Reviewer #2: Yes

5. Is the manuscript presented in an intelligible fashion and written in standard English?

Reviewer #1: Yes

Reviewer #2: Yes

6. Review Comments to the Author

Reviewer #1: All the suggested changes are incorporated and I recommend the manuscript for publication in PLOS ONE.

Reviewer #2: Congratulations to all the authors for conducting outstanding research on the gamma-ray mediated improvement of catalytic efficiency and thermostability of glucoamylase by substituting the active site leucine with isoleucine from Super Koji (Aspergillus oryzae). You have thoroughly addressed all the comments, and I believe publishing your work in PLOS ONE will allow it to reach a wide audience, benefiting many in the field.

7. PLOS authors have the option to publish the peer review history of their article (what does this mean? ). If published, this will include your full peer review and any attached files.

**Do you want your identity to be public for this peer review?** For information about this choice, including consent withdrawal, please see our Privacy Policy .

Reviewer #1: No

Reviewer #2: **Yes: ** Muhammad Tayyab Akhtar

---

## [Editor Report · Acceptance letter]

PONE-D-24-53783R1

PLOS ONE

Dear Dr. Ali,

I'm pleased to inform you that your manuscript has been deemed suitable for publication in PLOS ONE. Congratulations! Your manuscript is now being handed over to our production team.

Kind regards,

on behalf of

Dr. Habibullah Nadeem

Academic Editor

PLOS ONE